# Social vulnerability during the COVID-19 pandemic in Peru

**Carlos Orlando Zegarra Zamalloa** [1], **Pavel J. Contreras** [2]*, **Laura R. Orellana**[3], **Pedro Antonio Riega Lopez**[4], **Shailendra Prasad**[5], **María Sofía Cuba Fuentes**[2]

**1** Harvard T H Chan School of Public Health, Harvard University, MA, United States of America, **2** Centro de Investigación en Atención Primaria de Salud, Universidad Peruana Cayetano Heredia, Lima, Perú, **3** Emerge, Universidad Peruana Cayetano Heredia, Lima, Perú, **4** Facultad de Medicina, Universidad Nacional Mayor de San Marcos, Lima, Perú, **5** Center for Global Health and Social Responsibility, University of Minnesota, MN, United States of America

* pavelcontreras18@gmail.com, pavel.contreras.c@upch.pe

**Data Availability Statement:** All relevant data from our study are available in the Github repository (https://github.com/lorellanac/SVI-and-COVID-19-in-Peru.git).

## Abstract

The COVID-19 pandemic has demanded governments and diverse organizations to work on strategies to prepare and help communities. Increasing recognition of the importance of identifying vulnerable populations has raised a demand for better tools. One of these tools is the Social Vulnerability Index (SVI). The SVI was created in 2011 to identify and plan assistance for socially vulnerable populations during hazardous events, by providing disaster management personnel information to target specific areas. We aimed to evaluate and determine the social vulnerability in different provinces and districts of Peru in the context of the COVID-19 pandemic using an adapted version of the SVI index. Ecological, observational, and cross-sectional study was conducted. We adapted the SVI and collected indicators related to COVID-19. We organized and analyzed the population data of the 196 provinces of Peru, using data from government institutions. We found a distribution of high and very high SVI in the mountainous areas of Peru. High and very high social vulnerability indexes, due to the presence of some or all the variables were predominantly distributed in the provinces located in the southern and highlands of the country. The association between mortality rate and social SVI-COVID19 was inverse, the higher the vulnerability, the lower the mortality. Our results identify that the provinces with high and very high vulnerability indexes are mostly located in rural areas nearby the Andes Mountains, not having a direct correlation with COVID-19 mortality.

## I. Introduction

The COVID-19 pandemic has demanded governments and diverse organizations to work on strategies to prepare and help communities to face the pandemic. Peru has been one of the most affected countries in the world.

Certain populations could be more vulnerable to COVID-19 and more likely to manifest with more severe disease. This vulnerability reflects limited capacity of certain populations to mitigate, treat, and delay the transmission of pandemic disease, and reduce its clinical,

**Funding:** This project was funded by a Global Engagement Grant from the Center for Global Health and Social Responsibility, University of Minnesota (GEG to SP). Research reported in this publication was also supported by the Fogarty International Center of the National Institutes of Health under grants D43TW009345 awarded to the Northern Pacific Global Health Fellows Program and D43TW009763 awarded to the Program for Advanced ReseArch Capacities for AIDS in Peru (PARACAS) (FIC/NIH to PJC). The content is solely the responsibility of the authors and does not necessarily represent the official views of the National Institutes of Health. The funders had no role in study design, data collection and analysis, decision to publish, or preparation of the manuscript.

**Competing interests:** The authors have declared that no competing interests exist.

economic, and social impacts [1,2]. Some of the factors contributing to this vulnerability, particularly those from the social determinants of health, are preventable and mitigable.

Social vulnerability (SV) is described as "the properties of a person or group in terms of their ability to anticipate, deal, resist and recover from the impact" of a discrete and identifiable event in nature or society [3]. In addition, social vulnerability mainly refers to the resilience of communities when confronted by external stresses on human health, stresses such as natural or human-caused disasters, or disease outbreaks -such as COVID19 [4]. SV is closely related to the presence of social determinants of health. The World Health Organization defines these as "the circumstances in which people are born, grow, work, live and age, including the broader set of forces and systems that influence the conditions of life every day" [5]. These forces and systems include economic policies and systems, development programs, social norms and policies, and political systems [6,7].

Evidence indicates that people with more economical disadvantages are most vulnerable at all stages, before, during, and after a catastrophic event [8]. Socially vulnerable communities are more likely to experience high rates of mortality, morbidity, and property destruction, and are less able to achieve a post disaster recovery than less socially vulnerable communities [9]. These vulnerable communities include racial and ethnic minorities; children, the elderly or disabled people; and residents of certain types of housing, particularly high-rise or mobile houses [10].

Evidence suggests that the most vulnerable people or communities have not been sufficiently considered in the planning and response to adverse events by local and national authorities. For example, in the case of Peru, the COVID-19 related information may not adequately reach indigenous communities that speak native languages (Quechua, Aymara, or Amazonian languages) [11], and health services access for affected populations is difficult due to the absence of adequate roads or nearby health facilities [12].

These populations need to be identified in the preparation of local response planning by government agencies and non-governmental organizations. Effectively decreasing social vulnerability minimizes both human suffering and financial loss related to the provision of social services and public assistance after a disaster [9].

Within the territorial framework of Peru, local authorities could identify vulnerable communities in their geographical area, but Peruvian heath institutions are commonly underfunded, understaffed, and stretched thin by ongoing health and social service responsibilities would have a tough time to accomplish [13]. Increasing recognition of the importance of identifying vulnerable populations has raised a demand for better tools. One of these tools is the Social Vulnerability Index (SVI).

The SVI was created in 2011 by the Centers for Disease Control (CDC)/ Agency for Toxic Substances and Disease Registry's (ATSDR) Geospatial Research, Analysis & Services Program (GRASP) [14], to enable the public health community to quickly and precisely identify and plan assistance for socially vulnerable populations over the entire course of hazardous events [15]. The index was created to be applied during all phases of the emergency, including planning, response, mitigation, and recovery. It was first used retrospectively in the Hurricane Katrina disaster in New Orleans in 2005 where the potential predictive power of the SVI was studied [10]. The SVI can provide state, local, and tribal disaster management personnel information to target specific areas that may be socially vulnerable before, during, and after a hazard event.

The SVI has been validated and adapted in different parts of the world, such as Botswana [16] and China [17]. In the case of COVID19, it was used in Nigeria [18] and the United States, showing its adaptability to different contexts, and its ability to help authorities allocate adequate resources [19].

Potential uses of SVI are: help emergency response planners and public health officials identify and map the communities that will most likely need support before, during, and after a hazardous event and to estimate the number of needed supplies, or where to allocate and prioritize material and human resources, also it could help decide how many emergency personnel are required to assist people and to identify communities that will need continued support to recover following an emergency or natural disaster [20].

The COVID-19 pandemic caused a negative impact on the population's health and on their ability to deal with and recover. However, this vulnerability has not yet been characterized enough. Specifically, any measurement of social vulnerability needs to be considered in the specific context of vulnerability to COVID-19 [21].

This would lead to identifying the geographical regions that, due to their socioeconomic characteristics, infrastructure, population, and access to health, may be more vulnerable and require the greatest support and attention from the national or regional government.

## II. Methods

### Study design

Ecological, descriptive, observational, and cross-sectional.

### Population

People who reside in Peru as surveyed and recorded in the databases of the Peruvian Ministry of Health, the National Institute of Statistics and Informatics, and the National Census of 2017.

### Analysis unit

We analyzed provinces of Peru; provinces are intermediate territorial. Provinces can assume the role of basic unit of territorial development planning due to its intermediate nature. There are 196 provinces of Peru that can be classified by strata according to their number of inhabitants, which correlates directly with their human development index (HDI) [22]. The first stratum corresponds to Metropolitan Lima and Callao with more than 10 million inhabitants. The second stratum, with a population from 300 thousand to 1 million inhabitants, corresponds to 17 provinces. The third stratum with populations of 50,000 to 300,000 inhabitants comprises 102 provinces. Finally, stratum four, with populations of less than 50,000 inhabitants, corresponds to 75 provinces. Except for stratum 1, which only contains coastal provinces, all strata have coastal, mountain and jungle provinces.

### Data sources

The information was extracted from the virtual platforms of the National Institute of Statistics and Informatics (INEI) including data from the 2017 National Census (http://censo2017.inei.gob.pe/), of the Ministry of Transport and Communications, which contains the route classifier (https://portal.mtc.gob.pe/transportes/caminos/rutas.html), the National Health Superintendence (SUSALUD) http://datos.susalud.gob.pe/dataset/data-hist%C3%B3rich-of-the-register-of-daily-beds-available-and-occupied-of-the-format-f5002-v1, the HIS-MINSA administrative register (request for access to public information) for the estimation of morbidities except of HIV, the HIV Situation Room of the National Center for Epidemiology, Prevention and Control of Diseases of the MINSA (https://www.dge.gob.pe/vih/), the National Computerized System of Deaths (SINADEF) https://www.datosabiertos.gob.pe/dataset/informaci%C3%B3n-de-fallecidos-del-sistema-inform%C3%A1tico-nacional-de-defunciones-sinadef-ministerio. Except for HIS-MINSA, the data was publicly available.

INEI is the official government agency of Peru in charge of carrying out national censuses, the last of which was in 2017. Fourteen indicators were extracted from this source corresponding to the topics 'socioeconomic status' (05), 'household composition' and disability' (04), 'minority groups and language' (02), as well as 'type of housing and transportation' (03). SUSALUD regulates the collection, transfer, dissemination, and exchange of information generated or obtained by IAFAS, IPRESS and IPRESS Management Units, and publishes aggregated indicators of the country's health services supply capacity. The SINADEF is the computer application that allows the entry of data of the deceased, generation of the death certificate and the statistical report; includes general deaths, fetal deaths, and deaths of unidentified persons.

## Instrumentation- Social Vulnerability Index against COVID-19 (SVI-COVID19)

We adapted the SVI [14] to the context of COVID-19 by combining indicators related to the social determinants of health, which measure the expected negative impact of any kind of disasters, and two factors to determine vulnerability against COVID 19: risk factors associated with poor prognosis of the disease and access to health services.

A total of 33 variables distributed in six central themes were evaluated. Four themes correspond to the CDC SVI and refer to 1) socioeconomic status, 2) household composition and disability, 3) belonging to a minority group and language, and 4) type of housing and transportation. The COVID-19 specific factors were: 5) *epidemiological factors*, based on guidelines from the WHO and the Peruvian Ministry of Health, which identify populations at risk of acquiring severe forms of the disease and, therefore, have higher mortality from COVID-19; and 6) *health system factors*, including data related to the capacity, strength, and readiness of the health system in relation to the needs posed by COVID-19. We also included the number of hospital beds, total health spending and the percentage of regional health budget spending, number of doctors per 100,000, as well as the number of ICU beds and the number of mechanical ventilators available by region.

## Estimation of SVI-COVID19

For the construction of the SVI-COVID-19 indicators, the 33 variables were expressed in population percentages or incidents, which were arranged in ascending order per the 196 provinces of Peru. We ranked negative variables (i.e., poverty) from 1 to 196, and positive variables (i.e., income) whose higher percentage value in the population meant less vulnerability from 196 to 1. Subsequently, we calculated the percentile-ranks with the formula: Percentile-rank = (Rank—1) / 195. We then summed up the indicators by topics of the SVI-COVID19 and performed the calculation of ranks and percentile ranks by topic. Finally, the index was classified into quintiles, obtaining the following groups: very low vulnerability (<20%), low (20–40%), moderate (40–60%), high (60–80%), and very high (> 80%). We then calculated the death rate from COVID-19 per 100,000 inhabitants for each of the provinces of Peru between April 2020 to September 2021. This period includes both two more lethal COVID-19 waves during the pandemic in Peru.

Death from COVID-19 was determined under the standard established by SINADEF, which includes cases confirmed by a positive molecular, antigenic, or serological laboratory test, as well as the diagnoses indicated in the medical certificate.

## Statistical analysis

We performed descriptive statistics by calculating the means and medians as appropriate for each SVI-COVID19 indicator. We then created a bivariate map to show the vulnerability

index for each province and the COVID-19 mortality rate as a dependent variable. Additionally, we performed a multivariable analysis using a negative binomial regression. Incidence rate ratio (IRR) were calculated and adjusted by population density with their respective 95% confidence interval (95% CI). The bivariate map was generated in ArcGIS 10.7 and the statistical analysis was made in the STATA v.17.0 statistical software package.

### Ethical considerations

The research ethics committee of the Universidad Peruana Cayetano Heredia approved the study protocol with certificate No. 501-28-20. Since this was an ecological study with a secondary database, no individuals were evaluated and no personal data were accessed, so there were no risks associated with data confidentiality. In this sense, informed consent was not required for our study.

## III. Results

We evaluated all 196 provinces of Perú. The descriptive statistics of the 33 indicators by topic are shown in Table 1. Monthly per capita income is one of the indicators with a bigger range of inequality, averaging S/.913 ($234) in the least vulnerable provinces and S/.392 ($ 100) in the most vulnerable provinces. Likewise, the percentage of the indigenous population that do not speak Spanish in the most vulnerable provinces is three times the percentage of the least vulnerable provinces.

In epidemiological variables, we observed that the median of the immunocompromised rate, respiratory disease, and diabetes registers is higher among the less vulnerable population except for the median obesity rate which is higher in the most vulnerable population.

Among the health system province indicators: number of doctors, mechanical ventilators, hospitalizations, and ICU beds; the most remarkable difference was between provinces with higher and lower social vulnerability index.

Fig 1 shows the provinces of Peru according to the Social Vulnerability Index in COVID-19, with the most vulnerable provinces located in the central strip of the country and a slightly higher concentration among the southern regions and center. Additionally, the death rate from COVID-19 is classified in ranges of 200, with the highest range of 600–799 deaths per 10,000 inhabitants seen in the provinces of the central coastline and some provinces of the north and south coast. Figs 2–7 show the distribution of the SVI in Perú for each of its six themes.

The highest death rate from COVID-19 occurred in the provinces with moderate vulnerability with an average of 429 deaths per 100,000 inhabitants (SD: ± 209.69), the lowest rate was found in the highly vulnerable provinces with 254 deaths per 100,000 inhabitants (SD: ± 87.92). The provinces with very high vulnerability have a smaller population with 39,266 inhabitants on average.

In the regression analysis, we found that for each increase in the percentile of total social vulnerability, the risk of cumulative incidence of death from COVID-19 decreases by 29% (crude Relative Risk: 0.71, 95% CI: 0.55–0.92). When this is adjusted by population density it changes slightly, with the risk of cumulative incidence of death from COVID-19 decreasing by 28% (adjusted Relative Risk: 0.72 95% CI: 0.57–0.94), both statistically significant associations (p = 0.01).

When disaggregating the regression analysis by topic, most often we observed a decrease in the risk of the incidence of deaths from COVID-19 as the percentile of social vulnerability of the province increases. The relative risks are less than one in the other categories, from 0.30 concerning socioeconomic status to 0.73 in healthcare. In all of them, the results

**Table 1. Descriptive statistics for Peru provinces vulnerability indicators (n = 196).**

| Theme | Indicator | Social Vulnerability Index | | | | |
|---|---|---|---|---|---|---|
| | | Very Low | Low | Moderate | High | Very high |
| | Total Population thsnd(.)* | 69.4 (40.4–144.4) | 83.3 (49.6–144.2) | 80 (49.2–162) | 41.7 (20–66.8) | 30.4 (17.7–51.3) |
| Socioeconomic Status | % Persons in poverty† | - | - | - | - | - |
| | % Unemployed persons | 54.8 (5.6) | 58.2 (6.7) | 58.5 (6.2) | 64.3 (5.9) | 69.6 (6.7) |
| | Per Capita Monthly Income | 922.6 (339.8) | 739.2 (286.7) | 710.7 (219) | 527 (195.6) | 39168.5 (124) |
| | % Persons with no High School | 22.3 (9.1) | 26.1 (9.7) | 26.4 (12.6) | 31 (7.1) | 35.1 (4.5) |
| Household Composition and disability | % Persons aged 65 and older | 10.7 (2.3) | 11.8 (3.4) | 12.8 (4.1) | 15.7 (4) | 17.1 (4) |
| | % Persons aged 17 and younger | 37.7 (7.6) | 38 (7.3) | 37.4 (7.5) | 36 (5) | 36.6 (5) |
| | % Persons with a disability | 8.8 (1.6) | 9.2 (2.1) | 9.9 (2.4) | 10.9 (2.5) | 11.9 (2.9) |
| | % Single parent household | 18 (3.8) | 18.3 (4.1) | 19.8 (3.2) | 18.1 (3.3) | 17.4 (2.7) |
| Race Ethnicity Language | % Indigenous persons | 18.8 (19.4) | 29 (30.5) | 29.2 (28.5) | 56.6 (32.9) | 73.6 (21.9) |
| | % Persons who do not speak Spanish | 10.1 (11.4) | 18 (22.8) | 18.8 (22.1) | 40.8 (28.6) | 65.5 (22.3) |
| Housing Transportation | % Housing with poor structures | 36.8 (21.5) | 49.6 (24) | 48.1 (22.7) | 73 (18.2) | 81.2 (9.3) |
| | % Occupied housing units with more people than rooms | 14.2 (2.2) | 14.8 (2) | 15.2 (2.3) | 15.2 (2.8) | 15.1 (1.8) |
| | % Paved roads | 10.3 (19.1) | 10.1 (18.4) | 12.1 (18) | 14.3 (23.5) | 21.2 (33) |
| Epidemiological Factors | Cardiovascular Disease Rate per 1000 inh.* | 21 (14.9–26.9) | 23.3 (18.3–29.3) | 23.2 (17.6–29.5) | 24 (12.7–35.7) | 19.6 (10–34.4) |
| | Respiratory Disease Rate per 1000 inh.* | 185.8 (151.4–221.8) | 158.4 (140.5–215.6) | 169.4 (143.2–216.7) | 167.5 (148.2–213.2) | 170.1 (157.2–197.4) |
| | Persons Inmunocompromised Rate per 1000 inh.* | 1.6 (0.7–5.4) | 0.8 (0.4–2.8) | 2.4 (0.6–5.8) | 0.4 (0.2–1.1) | 0.2 (0–0.5) |
| | Obesity Rate per 1000 inh.* | 34.3 (21.4–60.3) | 45 (31.2–64.7) | 42.3 (32.9–53.8) | 59.1 (37.1–73) | 63.5 (46.9–84.1) |
| | Diabetes Rate per 1000 inh.* | 4.1 (1.8–7.2) | 4.3 (2.1–6.3) | 4.8 (2.6–8.6) | 1.9 (1.4–2.6) | 1.8 (0.9–3.1) |
| | Population Density* | 14.8 (4.5–48.3) | 25.8 (9.1–42.7) | 26 (10.3–65.6) | 10.3 (5.4–18.6) | 18.7 (9.5–26.2) |
| | Number of deaths from influenza and pneumonia per 100,000 inhab.* | 6.9 (0–13.9) | 5.2 (2.3–10.6) | 7.2 (1.7–11.7) | 4 (0–9) | 5.9 (3.1–10.2) |
| Healthcare System | Hospital beds per 100,000 inh* | 128.5 (91.3–199.1) | 137.4 (100.4–197) | 151.7 (110.1–188.5) | 146.4 (100–234.2) | 116.4 (85.2–155.4) |
| | Intensive Care Beds (ICU) per 100,000 inh.* | 3.6 (0–10.5) | 0 (0–7.5) | 5.1 (0–8.7) | 0 (0–2.2) | 0 (0–0) |
| | Number of physicians per 100,000 inh.* | 77.4 (51.2–115.7) | 75.2 (52–99.1) | 86.5 (66.5–102.4) | 84.2 (61.6–110.8) | 67.9 (60.1–87.7) |
| | Number of mechanical ventilations per 100,000 inh‡ | 16.3 (7.3) | 14.8 (7) | 14.5 (8.7) | 14.4 (6.8) | 10.3 (5.2) |
| | % Expenditure of the regional health budget | 93.2 (7.7) | 87.8 (15.1) | 86.9 (18.5) | 89.4 (17.1) | 79.6 (23.1) |
| | Per capita spending on health* | 27.2 (18.8–37.1) | 29.8 (22.1–37.1) | 28.2 (20.8–36.4) | 29.8 (20.2–45.3) | 24.9 (19.7–45.3) |
| | COVID-19 Health Labs per 100,000* | 3.2 (0.8–9.9) | 1.9 (0.8–4) | 1.8 (0.8–4.5) | 3.5 (1.4–7) | 2 (0–5) |
| | Servicios de emergencia por 100,000* | 2.9 (1.1–6.3) | 2.4 (1.1–6.8) | 1.7 (0.8–4.3) | 4.7 (1.3–8.7) | 5.8 (1.1–11.7) |
| Covid-19 Mortality Rate per 100,000 inh‡ | | 372.05 | 359.18 | 429.31 | 344.65 | 254.26 |
| | | ±185.24 | ±211.24 | ±209.69 | ±156.48 | ±87.92 |

†Information available in ranks

*Median (p25-75)

‡ Mean (SD).

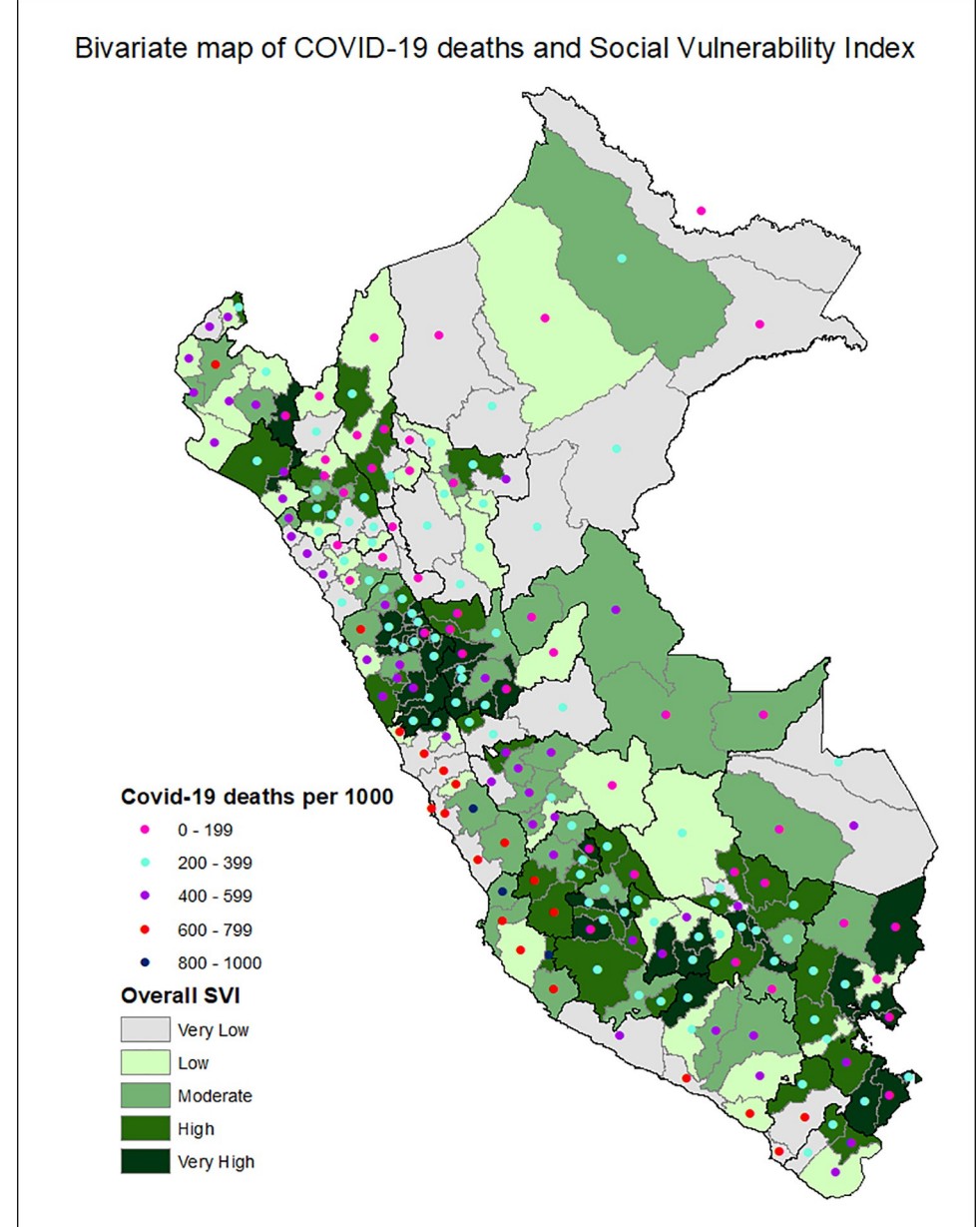

**Fig 1. Bivariate map of Social Vulnerability Index and death rate from COVID-19 per 10,000 inhabitants.**

in the adjusted model are statistically significant (p = 0.01). Except for categories 2 and 5 ("Household, composition and disability" and "Epidemiological Factors") where we see an opposite situation. For each increase in the percentile of these categories, the risk of cumulative incidence of death from COVID-19 increased by 84% (IRRa: 1.84 95% CI: 1.45–2.35) and 152% (IRRa: 2.52 95% CI: 2.03–3.13) respectively, after adjusting for population density (Table 2).

In all the regression analyses adjusted for population density, the directionality of the association is maintained.

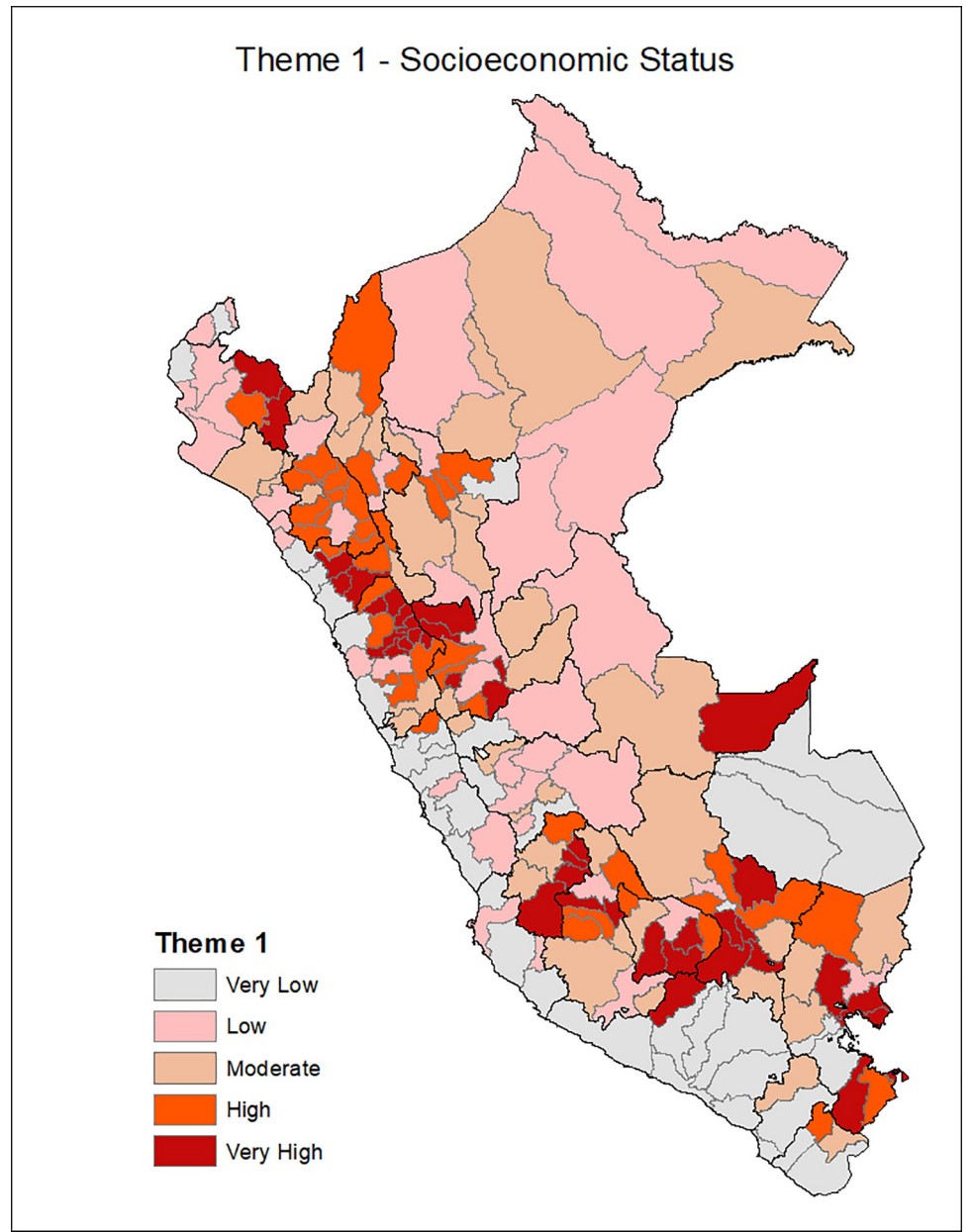

**Fig 2. SVI topic 1: Scioeconomic status.**

## IV. Discussion

In this study we adapted the SVI tool and included COVID-19 specific epidemiological (Obesity, Smoking, Cardiovascular diseases, Immunosuppression, etc.) and health system factors (ICU beds, number of physicians, per capita expenditure in health, number of COVID19 laboratories, etc.) to categorize the social vulnerability in various provinces of Peru. We then compared these to the mortality data from 2017 to 2020. As expected, we identified the distribution of the high and very high vulnerability index in rural provinces located or bordering the Andes Mountains. These are due to the lower access to health services and other public services, as well as unfavorable socioeconomic conditions in the populations in the mountainous regions

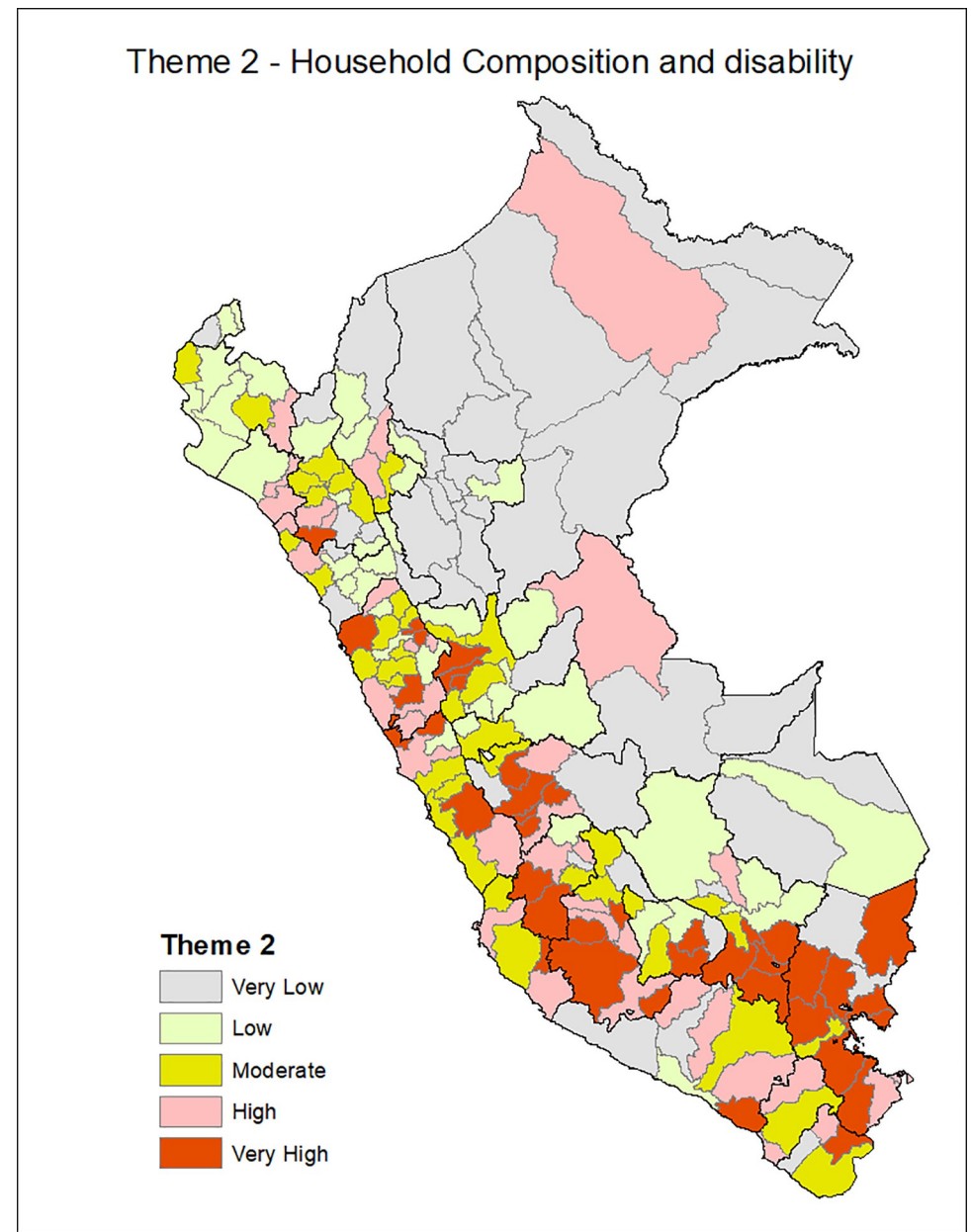

**Fig 3. SVI topic 2: Household composition and disability.**

of Peru [23]. The Ministry of Development and Social Inclusion of Peru also identifies these high Andean and Amazonian areas as the most vulnerable to any health condition or disaster [24]. Our vulnerability finding coincides and represents the unfavorable conditions of many provinces against COVID-19 and, probably, many other acute and chronic health problems.

The six themes in our instrument varied in distribution across Peru with socioeconomic vulnerability, housing, and transportation are distributed in the Peruvian highlands, the conditions of epidemiological vulnerability in the coastal areas, and the conditions of health system vulnerability in the southern part of the country.

We found that the mortality from COVID-19 increases with factors reflecting the composition of the household or the presence of a disability. Considering that the composition of the

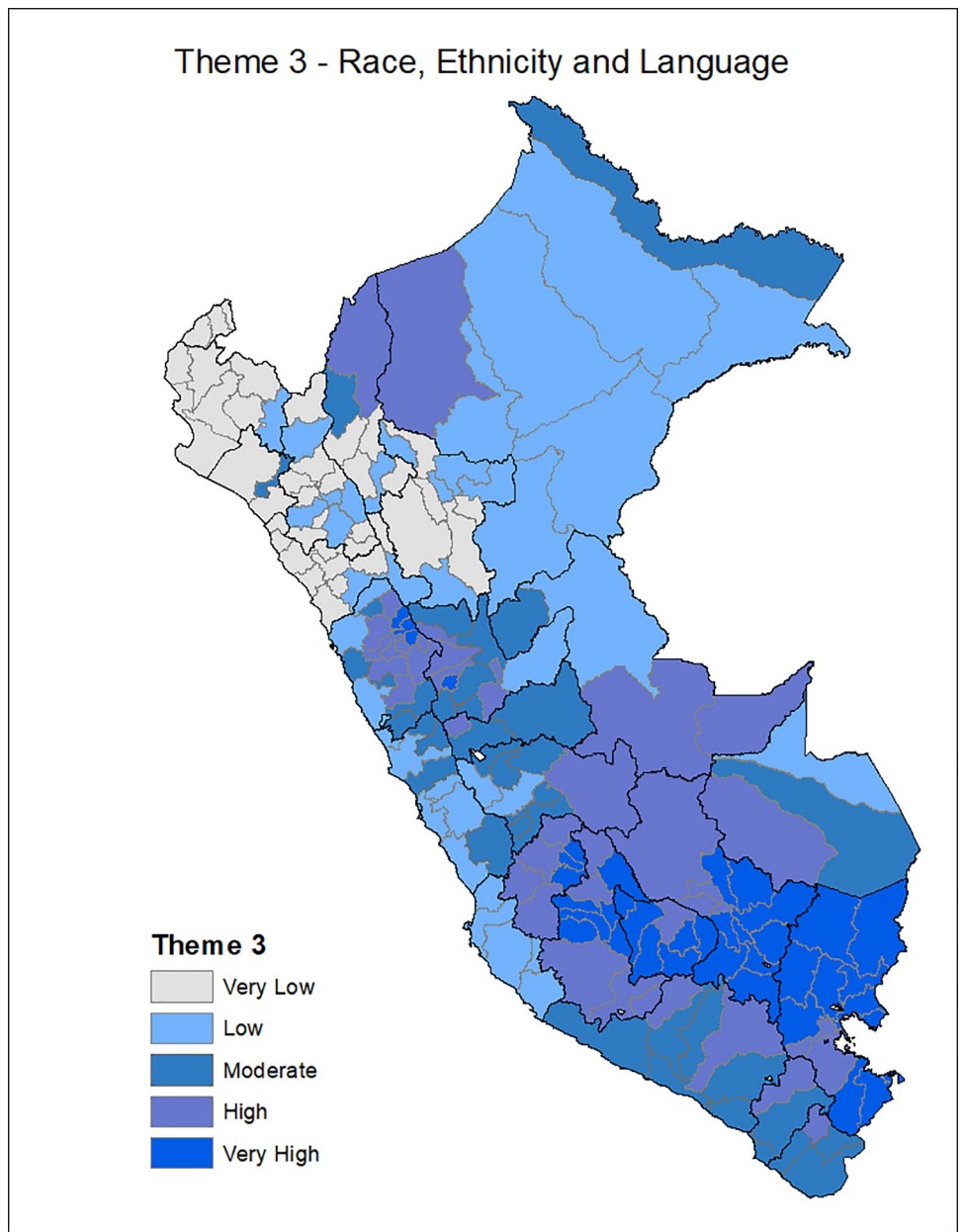

**Fig 4. SVI topic 3: Race/Ethnicity.**

household evaluates vulnerable groups such as the elderly, minors, and the presence of disability, this result reflects the known biological vulnerability to COVID-19 [25], or other factors that generate vulnerability in these groups. Similarly, and to a greater extent, we saw increased mortality from COVID-19 and the presence of epidemiological factors such as obesity, cardiometabolic, and chronic respiratory diseases, corroborating that the presence of this group of comorbidities increases the fatal outcome from COVID-19, therefore, placing populations with these comorbidities in a situation of higher mortality [26].

The death rate from COVID-19 was higher in coastal areas of Peru- areas in which the pandemic is known to have started and spread rapidly. However, in most of these provinces, the SVI-COVID19 assessment was low or very low, indicating that with the COVID-19 pandemic

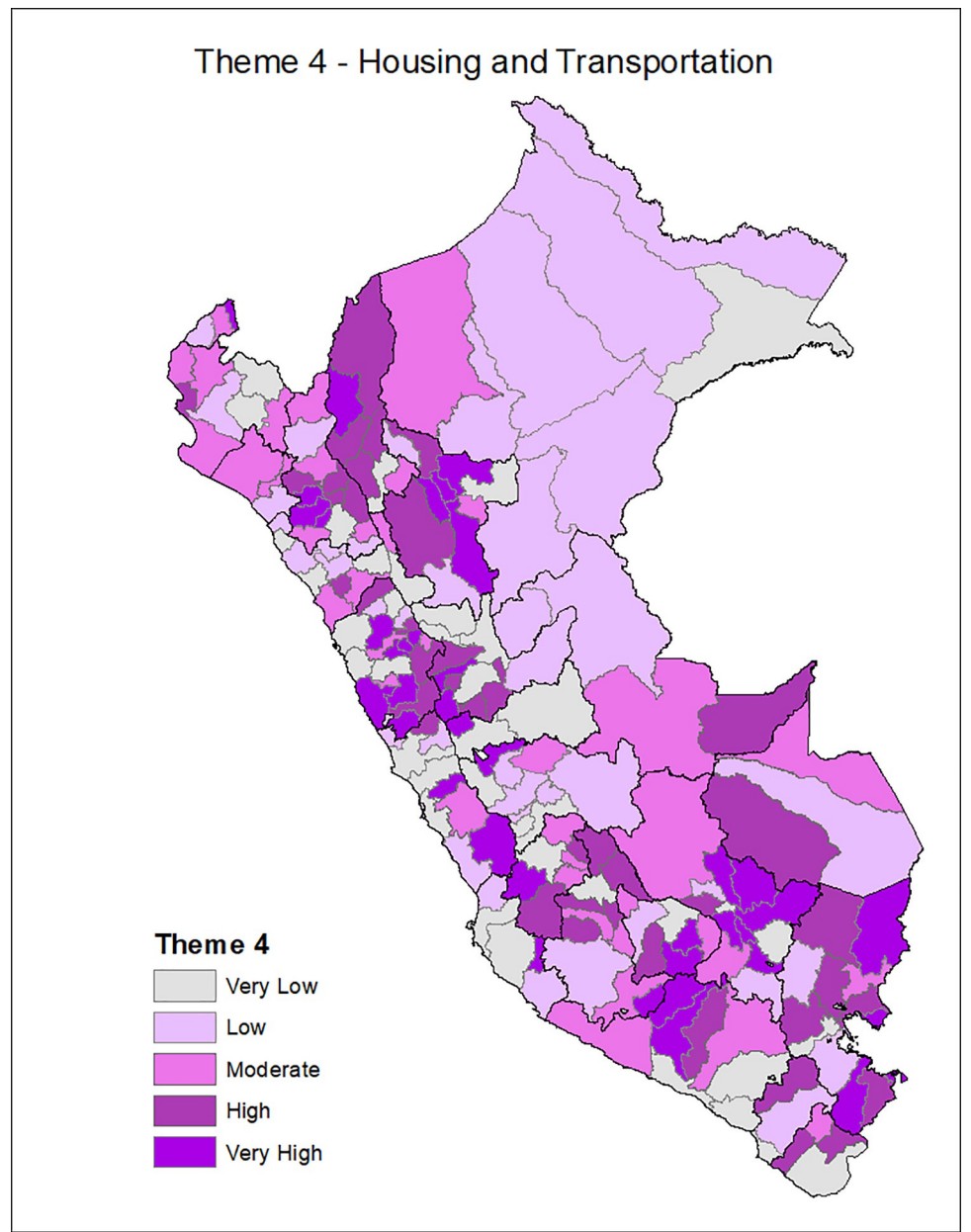

**Fig 5. SVI topic 4: Housing transportation.**

there are other variables or factors that should be considered beyond those included in our instrument. We can infer, for example, that coastal areas with low vulnerability have greater urbanization, higher population density that led to closer contact between individuals, greater saturation of hospitalization and emergency services in hospitals, higher mobilization of people within the city and between provinces, and a greater probability of areas of agglomeration thereby increasing the transmission of SARS- CoV-2 [27,28].

The set of the six themes evaluated with SVI-COVID19 present a negative association with the mortality rate from COVID-19, that is, the higher the social vulnerability, the lower the mortality. This result is partially consistent with studies carried out in the United States, where the evaluation of the first months of the pandemic mentions a positive association between

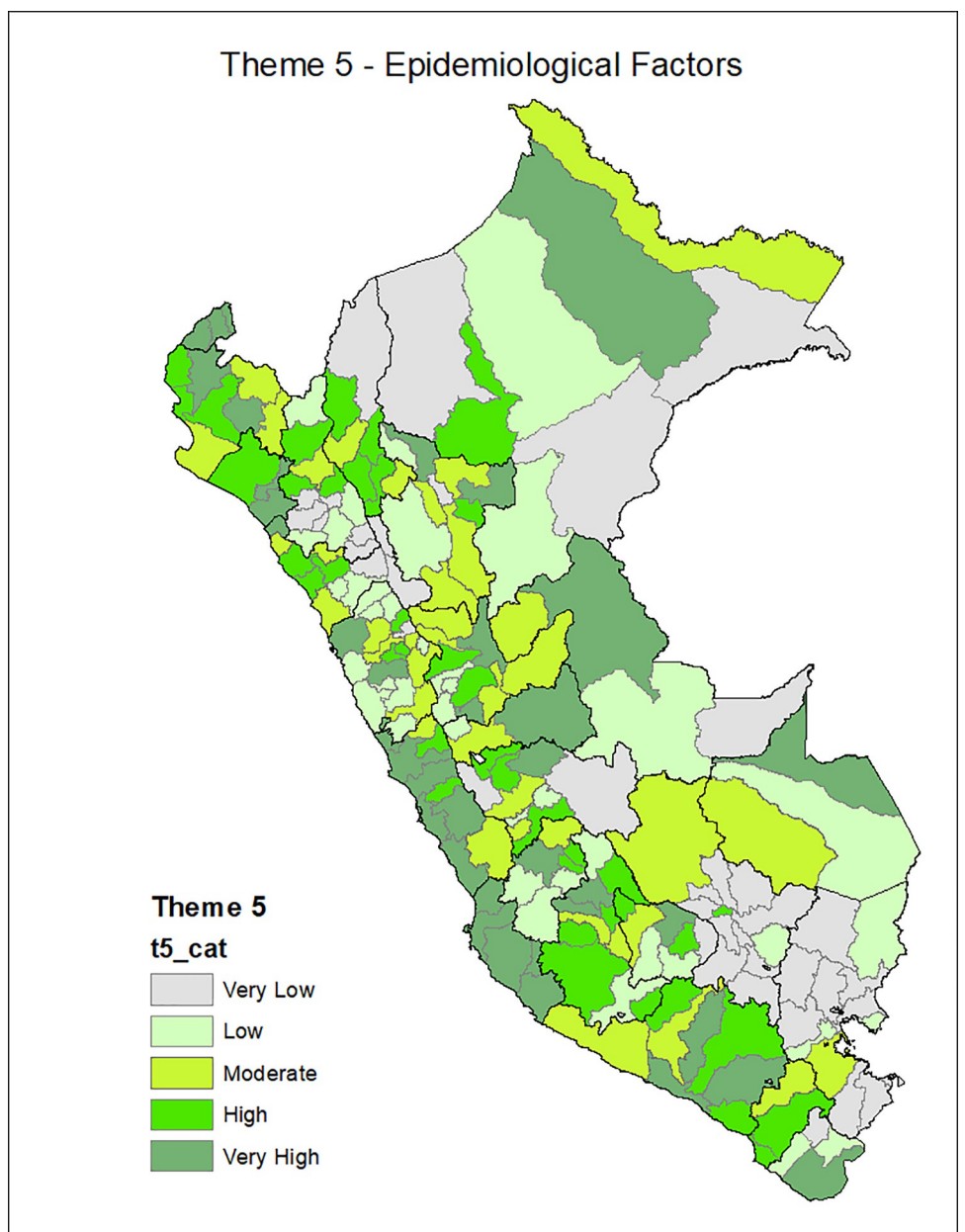

**Fig 6. SVI topic 5: Epidemiological factors.**

mortality from COVID-19 and SVI, which differs from our study. However, as the pandemic progresses, we observed a reverse association, showing that the relationship between both variables is not unidirectional, and is dependent on the time course of the pandemic [20,29]. This may be partly due to public health interventions. For example, since the start of the pandemic in Peru, there have been numerous interventions from the Ministry of Health at the national level to prevent and control the pandemic. Due to the emergency, these measures were multiple, and some were constantly variable, including implementation and relaxation periods, depending on the pandemic epidemiologic situation [30].

Two other elements can explain this factor. COVID-19 disproportionately impacted less vulnerable counties early and later in the pandemic. In the first place, the population with the

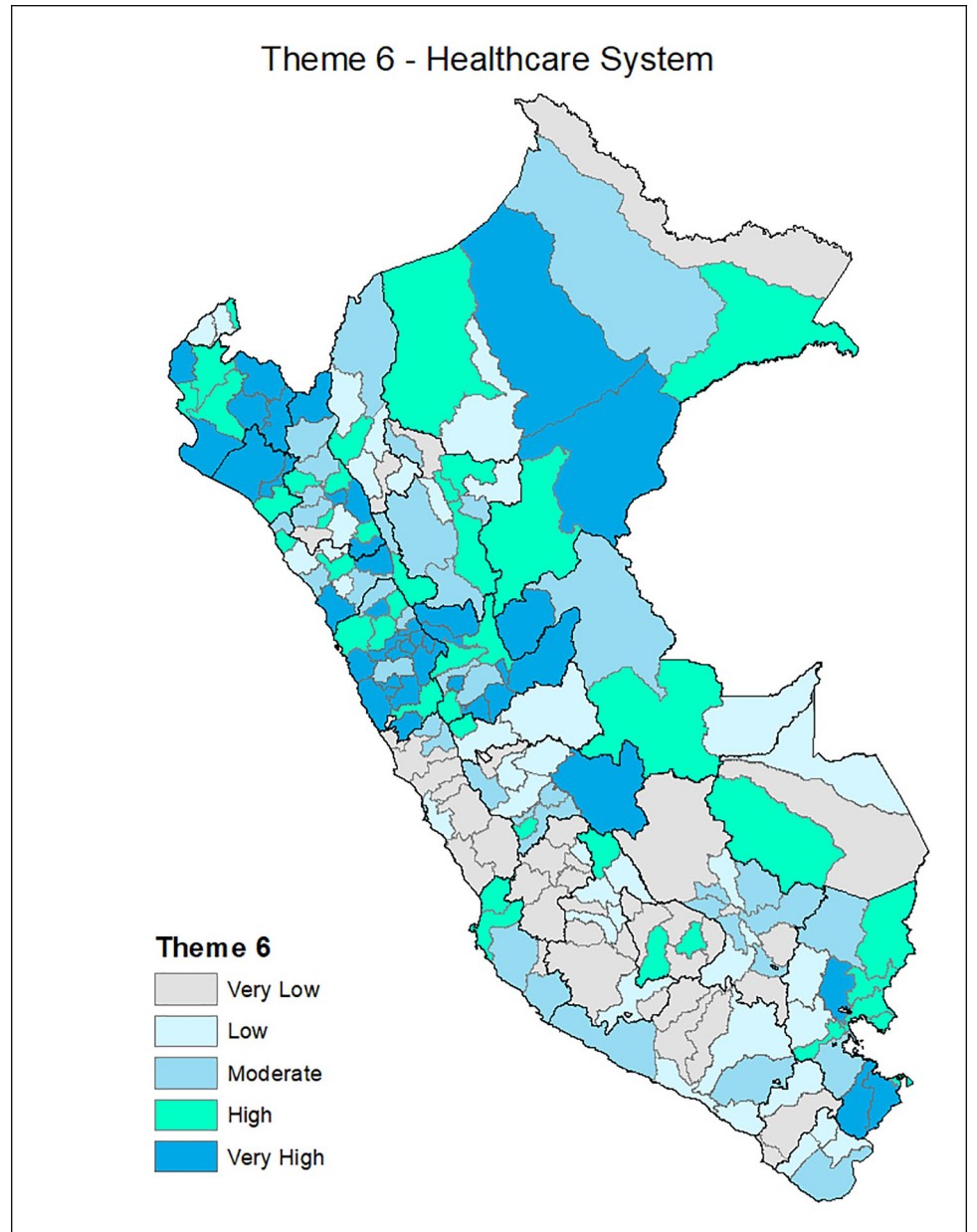

**Fig 7. SVI topic 6: Healthcare system.**

highest socioeconomic level was affected, later expanding to the rest of the population. This suggests that the impact of COVID-19 is not static but can migrate from less vulnerable counties to more vulnerable ones and back again over time [29].

At the beginning of the pandemic, the communities with the lowest vulnerability index had access to private health services, which to a certain extent were able to absorb the number of patients. However, with the general increase in cases, the health system (public and private) was saturated and unable to respond to the amount of population in need of medical care, affecting the entire population in general, and thus the "advantage" observed at the beginning, could have been diluted, generating in practice a barrier to access that could be comparable to that in which hospital services are non-existent or very scarce. It should be noted that at the

**Table 2. Incidence rate ratio of COVID-19 cumulative deaths from March 15, 2020 to September 31, 2021 among Peru's provinces (N = 196).**

| Themes | | Crude Model | | | Adjusted Model* | | |
|---|---|---|---|---|---|---|---|
| | | IRR | CI 95% | p | IRR | CI 95% | p |
| Theme 1 | Socioeconomic Status | 0.30 | 0.25–0.35 | <0.01 | 0.30 | 0.25–0.37 | <0.01 |
| Theme 2 | Household Composition and disability | 1.86 | 1.45–2.38 | <0.01 | 1.84 | 1.45–2.35 | <0.01 |
| Theme 3 | Race Ethnicity Language | 0.61 | 0.47–0.79 | <0.01 | 0.63 | 0.49–0.82 | <0.01 |
| Theme 4 | Housing Transportation | 0.61 | 0.48–0.77 | <0.01 | 0.63 | 0.50–0.80 | <0.01 |
| Theme 5 | Epidemiological Factors | 2.59 | 2.11–3.19 | <0.01 | 2.52 | 2.03–3.13 | <0.01 |
| Theme 6 | Healthcare System | 0.74 | 0.57–0.96 | 0.03 | 0.73 | 0.56–0.95 | 0.02 |
| SVI—CDC | | 0.54 | 0.43–0.68 | <0.01 | 0.57 | 0.45–0.72 | <0.01 |
| SVI—COVID-19 | | 0.71 | 0.55–0.92 | 0.01 | 0.72 | 0.57–0.94 | 0.01 |

*Adjusted by population density.

Incidence rate ratio (IRR) and 95% confidence intervals (CI95%) were estimated using negative binomial regression.

beginning of the pandemic (hospital capacity indicators compared to LATAM) and the closing of this gap was partial, late and transitory [31].

On the other hand, social vulnerability is directly related to the existence of gaps in the six topics of our instrument, which can also be analyzed from the point of view of inequities and inequality in health, the latter understood as the different distribution of wealth and access to care according to geographic location. Thus, we consider that the SVI represents a measurement of health inequities, therefore, of the existing gaps in Peru [32,33].

It is necessary to consider the possible influence of social vulnerability with the underreporting of deaths, considering that SINADEF is a computer tool whose registration is more complex in areas without internet access, which has been exacerbated during the first and second waves of the pandemic [34].

The results of the negative association between SVI-COVID19 and the COVID-19 mortality rate could be explained in part by the low registration of mortality, specifically in populations with the greatest social vulnerability. The mortality registration in regions of Amazonas, Ayacucho, Cajamarca, Junín, Loreto, and Pasco, of Peru is less than or equal to 60%. Since these are regions with higher social vulnerability, our results may be skewed due to unregistered deaths.

During 2020 and 2021, registers showed a significant exceed in the number of deaths, according to the reference pattern previously estimated by the INEI for those years. The estimates of deaths expected by the INEI only reach the regional level, but not provinces or districts.

A limitation in our study is the use of mortality data. While the current mode of collecting mortality data by the internet has increased the overall collection of data, it has likely led to underreporting from areas with poor technology and internet connectivity [34].

The registration of deaths with medical certification in Peru–with documentation of the cause of death—in 2019 was 71.6%, so 54,372 deaths in 2019 did not have a medical death certificate or whose medical certificate of death was not entered into SINADEF or the EsSalud mortality database. SINADEF is not the only death registration system with medical certification of causes of death. EsSalud registered 17,483 deaths that are not registered in SINADEF. The registered deaths drop to 62.15% of all deaths if we consider deaths registered by SINADEF [34].

Additionally, other studies report that good socioeconomic conditions and health expenditures, parameters considered in the SVI, can reduce the rate of COVID-19 infection when the incidence is low, but not when the spread has an exponential advance [32].

Among the limitations of our study is the ecological design, since the results can only be interpreted at a population level and cannot address the characteristics that can help to identify the individual factors that can define the personal resilience ability to respond to the COVID19 pandemic. The data obtained from institutional databases could present underreporting bias or misclassification, especially for populations with less accessibility. The cross-sectional characteristic of our study limits the understanding of the evolutionary process of the risks or vulnerable conditions in relation to the health measures or policies assumed by the Government. Considering the existing inequalities within the 25 regions that make up Peru, further studies that look at vulnerability analysis with a greater spatial disaggregation, particularly for non COVID19 specific issues, including that of other diseases and conditions that have been unfortunately neglected due to the pandemic.

## V. Conclusions

Our results identify the provinces of the country with high and very high vulnerability, most of which are in the rural areas bordering the Andes Mountains. Through this mapping, the index can help government authorities to improve prevention, control, and resource management policies in more detail [20].

It is possible that this negative association found between SVI and the mortality rate could be related to the low reporting coverage of deaths in general and deaths from COVID-19 in the provinces with a higher index of social vulnerability.

Likewise, this study highlights the need for better registration and epidemiological reporting systems. As we have seen, there is a limitation, which mainly affects rural and poor populations, in terms of adequate health data, which can lead to misreporting and potentially increase health inequalities to the detriment of vulnerable people, especially in Andean and Amazonian areas. Likewise, it can orient authorities incorrectly by giving a false perception about the actual health needs of the populations under study and directing the limited resources oriented towards health.

Additional studies are required to determine the causes of the paradoxical results found in this study to assess the importance of other individual and community factors that may have affected the burden of disease due to COVID19.

## Acknowledgments

Dr. Javier Vargas Herrera for his contribution to the understanding of SINADEF.

## Author Contributions

**Conceptualization:** Carlos Orlando Zegarra Zamalloa, Shailendra Prasad, María Sofía Cuba Fuentes.

**Data curation:** Laura R. Orellana.

**Formal analysis:** Laura R. Orellana.

**Funding acquisition:** Shailendra Prasad.

**Investigation:** Carlos Orlando Zegarra Zamalloa, Pavel J. Contreras, María Sofía Cuba Fuentes.

**Methodology:** Carlos Orlando Zegarra Zamalloa, Pavel J. Contreras, Pedro Antonio Riega Lopez, Shailendra Prasad, María Sofía Cuba Fuentes.

**Project administration:** Pavel J. Contreras, María Sofía Cuba Fuentes.

**Software:** Laura R. Orellana.

**Supervision:** Carlos Orlando Zegarra Zamalloa, Pavel J. Contreras, María Sofía Cuba Fuentes.

**Writing – original draft:** Carlos Orlando Zegarra Zamalloa, Pavel J. Contreras, Laura R. Orellana, Pedro Antonio Riega Lopez, Shailendra Prasad, María Sofía Cuba Fuentes.

**Writing – review & editing:** Carlos Orlando Zegarra Zamalloa, Pavel J. Contreras, Pedro Antonio Riega Lopez, Shailendra Prasad, María Sofía Cuba Fuentes.

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
