## [Decision Letter · Decision Letter 0]

27 Jul 2022

PGPH-D-22-00628

Social Vulnerability during the COVID-19 Pandemic in Peru

Dear Dr. Contreras,

Thank you for submitting your manuscript to PLOS Global Public Health. After careful consideration, we feel that it has merit but does not fully meet PLOS Global Public Health’s publication criteria as it currently stands. Therefore, we invite you to submit a revised version of the manuscript that addresses the points raised during the review process.

The manuscript has been evaluated by two reviewers, and their comments are available below.

The reviewers have raised a number of major concerns. They request improvements to the reporting of methodological aspects of the study, for example, and they have several notes regarding the clarity of the manuscript.

Could you please carefully revise the manuscript to address all comments raised?

We look forward to receiving your revised manuscript.

Kind regards,

Julia Robinson

Executive Editor

Journal Requirements:

1. Please amend your detailed online Financial Disclosure statement. This is published with the article. It must therefore be completed in full sentences and contain the exact wording you wish to be published.

2. Please ensure that the funders and grant numbers match between the Financial Disclosure field and the Funding Information tab in your submission form. Note that the funders must be provided in the same order in both places as well.

3. Please send a completed 'Competing Interests' statement, including any COIs declared by your co-authors. If you have no competing interests to declare, please state “The authors have declared that no competing interests exist”. Otherwise please declare all competing interests beginning with the statement “I have read the journal's policy and the authors of this manuscript have the following competing interests:”

For more information, please see our guidelines: https://journals.plos.org/globalpublichealth/s/submission-guidelines#loc-competing-interests

4. All figures and supporting information files will be published under the Creative Commons Attribution License (creativecommons.org/licenses/by/4.0/). Authors retain ownership of the copyright for their article and are responsible for third-party content used in the article. 

Figures 1, 2, 3, 4, 5, 6, and 7: please (a) provide a direct link to the base layer of the map used and ensure this is also included in the figure legend; (b) provide a link to the terms of use / license information for the base layer. We cannot publish proprietary or copyrighted maps (e.g. Google Maps, Mapquest) and the terms of use for your map base layer must be compatible with our CC-BY 4.0 license. 

Please upload any written confirmation as an 'Other' file type. It must clarify that the copyright holder understands and agrees to the terms of the CC BY 4.0 license; general permission forms that do not specify permission to publish under the CC BY 4.0 will not be accepted. Note that uploading an email confirmation is acceptable.

Additional Editor Comments (if provided):

Reviewers' comments:

Reviewer's Responses to Questions

**Comments to the Author**

1. Does this manuscript meet PLOS Global Public Health’s publication criteria? Is the manuscript technically sound, and do the data support the conclusions? The manuscript must describe methodologically and ethically rigorous research with conclusions that are appropriately drawn based on the data presented.

Reviewer #1: Yes

Reviewer #2: No

2. Has the statistical analysis been performed appropriately and rigorously?

Reviewer #1: Yes

Reviewer #2: No

3. Have the authors made all data underlying the findings in their manuscript fully available (please refer to the Data Availability Statement at the start of the manuscript PDF file)?

Reviewer #1: Yes

Reviewer #2: Yes

4. Is the manuscript presented in an intelligible fashion and written in standard English?

Reviewer #1: Yes

Reviewer #2: No

5. Review Comments to the Author

Reviewer #1: Nice manuscript.

Authors need to take note of the following:

1. the abstract does not well represent the body of the work.

2. Line 68: remove the underscore sign

3. Line 70-71: the sentence is incomplete

4. Line 272-274: the citation appears counterintuitive. Are the authors sure of the information presented? Please verify

5. Can the 33 variables be presented in tabular format for the sake of clarity?

6. Please provide justification for the choice of study variables.

Reviewer #2: This study aims to investigate the relationship between SVI-COVID factors and COVID mortality in 196 provinces in Peru. My comments/questions are listed below. Overall, the manuscript does not meet the standard of publications and requires substantial revision.

1. I feel that Table 1 about SVI uses in Introduction is not needed and many are not relevant to this manuscript. The author may want to consider summarizing SVI in text.

2. Some sentences in Introduction do not provide meaningful messages. More in-depth literature review may be necessary. For example, page 6, line 112-113. However, SVI is expected to be different across provinces in any country. The extent of difference may be large or small. Social determinants are known to affect people and communities (this is very broad).

3. Please provide more details in data sources, in addition to listing the websites. For example, number of provinces (regions in Peru), brief summary of data. For the mortality data, why was April 2020 to Sep 2021 selected? Please also describe definition and approaches to identify death from COVID-19 in this section. I’m wondering if there is any difference in mortality over time?

4. Page 7, line 141. The reference is not related to COVID-19. Suggest cite the reference right after SVI.

5. In terms of estimation of SVI-COVID19, the variables were treated with equal weight. This may not be the most efficient approach. Ranking may result in some loss of information.

6. Did the author mean to say random “intercept”? From my understanding, there were a total of 196 observations (provinces). Random intercept is not feasible in this case. What does “multiple analyses” mean? Replications or change of methods? Please list any covariates here as well.

7. Please spell out all abbreviations the first time using it, e.g., CI, RR, IRRa. IRRa was used in text, whereas RR was used in Table 3. I think they refer to the same numbers? Please confirm what statistics to report and be consistent.

8. In table 2, should most of the numbers multiply by 100 since many units are in %? Please present lower and upper quartiles if choosing to report median and SD in the table.

9. In Discussion, the authors said that they adapted a validated tool to include COVID-19 specific factors to categorize the social vulnerability in various provinces of Peru (page 13, line 242). I don’t see where a validated tool was mentioned. Does this mean SVI? I suggest to say SVI directly. What are COVID-10 specific factors? Perhaps this means COVID-19 labs? Please clarify.

6. PLOS authors have the option to publish the peer review history of their article (what does this mean?). If published, this will include your full peer review and any attached files.

**Do you want your identity to be public for this peer review?** For information about this choice, including consent withdrawal, please see our Privacy Policy.

Reviewer #1: No

Reviewer #2: No

---

## [Decision Letter · Decision Letter 1]

7 Nov 2022

Social Vulnerability during the COVID-19 Pandemic in Peru

PGPH-D-22-00628R1

Dear Dr. Contreras,

We are pleased to inform you that your manuscript 'Social Vulnerability during the COVID-19 Pandemic in Peru' has been provisionally accepted for publication in PLOS Global Public Health.

Best regards,

Julia Robinson

Executive Editor

Reviewer Comments (if any, and for reference):

Reviewer's Responses to Questions

**Comments to the Author**

1. If the authors have adequately addressed your comments raised in a previous round of review and you feel that this manuscript is now acceptable for publication, you may indicate that here to bypass the “Comments to the Author” section, enter your conflict of interest statement in the “Confidential to Editor” section, and submit your "Accept" recommendation.

Reviewer #2: All comments have been addressed

2. Does this manuscript meet PLOS Global Public Health’s publication criteria? Is the manuscript technically sound, and do the data support the conclusions? The manuscript must describe methodologically and ethically rigorous research with conclusions that are appropriately drawn based on the data presented.

Reviewer #2: (No Response)

3. Has the statistical analysis been performed appropriately and rigorously?

Reviewer #2: Yes

4. Have the authors made all data underlying the findings in their manuscript fully available (please refer to the Data Availability Statement at the start of the manuscript PDF file)?

Reviewer #2: Yes

5. Is the manuscript presented in an intelligible fashion and written in standard English?

Reviewer #2: Yes

6. Review Comments to the Author

Reviewer #2: thanks for your response.

7. PLOS authors have the option to publish the peer review history of their article (what does this mean?). If published, this will include your full peer review and any attached files.

**Do you want your identity to be public for this peer review?** For information about this choice, including consent withdrawal, please see our Privacy Policy.

Reviewer #2: No
